# PLINDER:
# The protein-ligand interactions dataset and evaluation resource

**Janani Durairaj** [* 1 2]  **Yusuf Adeshina** [* 3]  **Zhonglin Cao** [4]  **Xuejin Zhang** [3]  **Vladas Oleinikovas** [3]
**Thomas Duignan** [3]  **Zachary McClure** [4]  **Xavier Robin** [1 2]  **Emanuele Rossi** [3]  **Guoqing Zhou** [4]  **Srimukh Veccham** [4]
**Clemens Isert** [3]  **Yuxing Peng** [4]  **Prabindh Sundareson** [4]  **Mehmet Akdel** [3]  **Gabriele Corso** [5]  **Hannes Stärk** [5]
**Zachary Carpenter** [3]  **Michael Bronstein** [3 6]  **Emine Kucukbenli** [4]  **Torsten Schwede** [1 2]  **Luca Naef** [3]

## Abstract

Protein-ligand interactions (PLI) are foundational to small molecule drug design. With computational methods striving towards experimental accuracy, there is a critical demand for a well-curated and diverse PLI dataset. Existing datasets are often limited in size and diversity, and commonly used evaluation sets suffer from training information leakage, hindering the realistic assessment of method generalization capabilities. To address these shortcomings, we present PLINDER, the largest and most annotated dataset to date, comprising 449,383 PLI systems, each with over 500 annotations, similarity metrics at protein, pocket, interaction and ligand levels, and paired unbound (*apo*) and predicted structures. We propose an approach to generate training and evaluation splits that minimizes task-specific leakage and maximizes test set quality, and compare the resulting performance of DiffDock when retrained with different kinds of splits.

## 1. Introduction

The protein-ligand field has seen a surge in the application of deep learning-based prediction methods, notably in tasks such as **rigid body docking** (Stärk et al., 2022; Lu et al., 2022; Corso et al., 2023) where the pose of a ligand within a given rigid protein structure is predicted, **flexible pocket docking** (Plainer et al., 2023; Qiao et al., 2024) which allows side-chain movements of pocket residues, **co-folding** (Qiao et al., 2024; Krishna et al., 2024; Abramson et al.,

2024) where both the protein conformation and ligand pose are predicted at once, **pocket-conditioned ligand generation** (Schneuing et al., 2023) where novel ligand molecules are generated within a given protein structure and pocket, **ligand-conditioned protein engineering** (Dauparas et al., 2023) where, conversely, novel protein sequences are designed to selectively bind a ligand, and **molecular scaffolding** (Chawdhury et al., 2021) where ligands are modified to enhance their affinity to a protein or pocket.

These methods hold promise in accelerating drug discovery and protein engineering by facilitating the accurate prediction of ligand binding poses within protein structures. However, their effectiveness relies heavily on the datasets used for training and evaluation, where several key considerations must be addressed: **1. Training set diversity** to move to high data regimes and learn the underlying patterns instead of simple memorization; **2. Low information leakage between train and test** to accurately assess generalisation capabilities and avoid inflating expected performance by over-fitting; **3. Test set quality** to avoid comparing prediction results to unreliable ground truth caused by varying experimental quality or missing atoms in and around the binding site; **4. Test set diversity** to showcase performance across a range of complex types and use-cases; and **5. Realistic inference scenarios** to move beyond the "re-docking" use-case where the ligand pose is predicted within the never-available experimental ligand-bound receptor conformation.

Despite the availability of numerous publicly reported protein-ligand interaction (PLI) structural datasets, many fall short in meeting these critical considerations. For instance, BioLip2 (Zhang et al., 2024), a sizable dataset, primarily emphasizes functional annotation and lacks suitable partitioning for training or testing machine learning-based methods. Other datasets such as PDBBind (Wang et al., 2005) and variants of it come with suggested splits, yet they are small and contains information leakage between data partitions. Attempts to address leakage issues in PDBBind remain limited in dataset size and do not provide principled evaluation of different leakage metrics chosen via model

---

[*]Equal contribution  [1]Biozentrum, University of Basel, Basel, Switzerland  [2]SIB: Swiss Institute of Bioinformatics, Basel, Switzerland  [3]VantAI, New York, USA  [4]NVIDIA  [5]MIT CSAIL  [6]Oxford University. Correspondence to: Janani Durairaj <janani.durairaj@unibas.ch>, Yusuf Adeshina <yusuf@vant.ai>.

*Accepted at the 1st Machine Learning for Life and Material Sciences Workshop at ICML 2024.* Copyright 2024 by the author(s).

retraining (Li et al., 2024). Selectively combining previously curated datasets using evolutionary classification of protein domains (ECOD) (Cheng et al., 2014) annotations, as done in DockGen (Corso et al., 2024), ensures novel test domains to assess generalization potential, but is still limited in dataset size, constrained by manual curation biases in ECOD domain annotation, and has limitations in assessing novel ligands and binding modes for shared ECOD domains.

PLINDER offers the largest and most diverse dataset of protein-ligand complexes, encompassing various types such as multi-ligand systems, oligonucleotides, peptides, and saccharides. It calculates similarity between complexes at the protein, pocket, PLI, and ligand levels, enabling the measurement of diversity and detection of information leakage. Additionally, PLINDER annotates complexes for quality and domain information, and proposes an approach to prioritize a diverse, high-quality test set with minimal leakage. It also links *holo* complexes to relevant *apo* and predicted structures, facilitating realistic inference scenarios. Through these measures, PLINDER aims to provide researchers in the protein-ligand field with a robust and reliable dataset for training and evaluating deep learning-based prediction methods, ultimately advancing the development of novel drug discovery and protein engineering approaches.

## 2. Method Overview

### 2.1. Dataset Curation and Annotation

We obtained all entries from the Protein Data Bank (PDB) (Berman et al., 2000) as of 2024-04-09, using MMCIF files in the PDB NextGen Archive resource (Choudhary et al., 2023). For entries solved by X-ray crystallography, we extracted entry and residue-level information from the corresponding X-ray validation reports. We generated each biounit assembly of each PDB entry using OpenStructure (Biasini et al., 2013), and obtained all protein-ligand interactions detected by the protein-ligand interaction profiler (PLIP) (Adasme et al., 2021) for all ligand-like chains. Only interactions between protein and ligand atoms or ligand atoms and water molecules are considered. A chain within a PDB entry was labelled as a ligand chain if one of the following was true: (1) the chain type was non-polymer, (2) there was a Biologically Interesting Molecule Reference Dictionary (BIRD) (Dutta et al., 2014) identifier associated with that chain, (3) the chain was of type polypeptide, oligosaccharide, or oligonucleotide and had less than 10 residues, (4) the chain was of type polypeptide and had less than 20 residues and no UniProt ID associated with it. Ligand chains within 4 Å of each other or having a detectable PLIP interaction are merged into the same PLI system. Each system is also associated with its "interacting" residues, consisting of residues participating in a PLIP interaction with the ligand chains of that system, and "neighboring" residues, consist-

ing of residues within 6 Å of the ligand, together making up the system pocket. Thus, PLI systems are defined by the combination of a PDB ID, biounit identifier, specific instance(s) of the ligand chain(s), and specific instance(s) of the interacting protein chain(s) (Figure 1A).

For each system, we provide annotations broadly categorised in Table A. All domain annotations are mapped to pocket residues, and the domain sharing the highest overlap with the pocket is retained as the pocket domain. Systems are labelled as *holo*, artifact or ion and annotated with ligand definitions and properties as described in Appendix A.1. We performed extensive molecule processing and cleaning efforts to correct bond, valence and chirality issues to ensure that the vast majority of PLINDER can readily be processed and used by the typical pre-processing and feature extraction routines employed by deep learning methods, and those which cannot are clearly annotated.

### 2.2. Protein-ligand system similarity

To cluster and split our curated dataset, we calculated protein and pocket similarity using MMSeqs (Steinegger & Söding, 2017) and Foldseek (van Kempen et al., 2024) (Figure 1B), PLI similarity using the alignments combined with unique PLIP interactions, and ligand similarity using ECFP4 fingerprints. MMSeqs and Foldseek searches were conducted on all PDB chains within all systems (E-value < 0.01, min_seq_id 0.2, max_seqs 5000) to yield alignments and query coverages. Protein similarities are determined based on lDDT score (only for Foldseek), identity percentage, sequence similarity, and query coverage-adjusted global versions of these scores. Per-residue alignment information was used to calculate pocket-level scores. PLIP interactions were made unique using the interaction type and type-specific attributes as defined in Table A9, and combined with pocket information in a weighted Jaccard similarity metric to compute a PLI-level similarity measure. See Appendix B for more details.

For each similarity metric and given thresholds (50, 70, 95, 100), a graph is created with systems as nodes and edges between systems which have a similarity score above the threshold for the specified metric. Note that all similarity values are adapted to be in the 0-100 range. Strong and weakly connected components, as well as communities based on the Parallel Louvain Method are identified for this graph, using NetworKit (Angriman et al., 2023), and systems within each cluster are labelled with the corresponding cluster identifier, adding additional similarity-centric annotations to each system.

### 2.3. Train-validation-test dataset splitting

We used Algorithm 1 to split the dataset into proto-train and test sets, to ensure that our requirements for training set

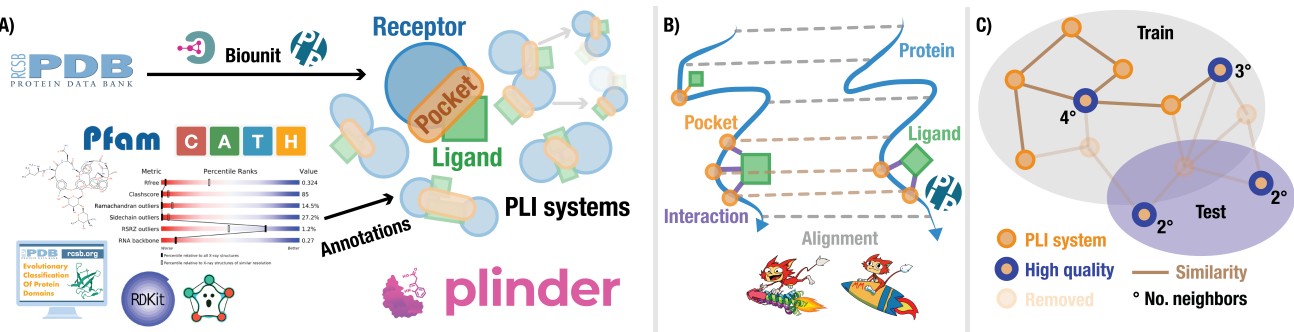

*Figure 1.* **A)** Curation and annotation of PLI systems from the PDB, **B)** Measuring similarity between PLI systems **C)** Splitting into diverse train and high quality test sets.

---

**Algorithm 1** Splitting

1: **Input:** systems $S$, clusters $C$, graphs $G$, depths $D$, maximum leakage count $M$, minimum cluster size $m$, number of representatives from each cluster $n$
2: **Output:** $proto\_train$ and $test$ systems
3: Initialize $proto\_test \leftarrow \emptyset$
4: Initialize $\forall s \in S, N_s = \emptyset$
5: **for** $s \in S$ **do**
6:    **if** $pass\_quality(s)$ **then**
7:       **for** $g = 1$ to $|G|$ **do**
8:          $N_s \leftarrow N_s \bigcup neighbors\_upto\_depth(s, G_g, D_g)$
9:       **if** $m < |N_s| < M$ **then**
10:          $proto\_test.insert(s)$
11: Sort $s \in proto\_test$ by $|N_s|$
12: Initialize $test \leftarrow \emptyset$
13: **for** $c \in C$ **do**
14:    $test \leftarrow$ up to $n$ from $proto\_test$ for $c$
15: Initialize $proto\_train \leftarrow S$
16: **for** $s \in test$ **do**
17:    $proto\_train \leftarrow proto\_train \setminus N_s$

---

size, test set quality, test set diversity, and low information leakage are met (Figure 1C). It is important to note that these considerations vary for different prediction tasks. For instance, in the case of rigid body docking methods, having a similar protein in train and test may not be considered leakage if the binding pocket location, conformation and/or pocket interactions with a ligand are sufficiently different. This allows us to evaluate the common use-case of docking a different ligand into a known drug target. However, for co-folding tasks, having such train-test pairs could potentially overestimate model performance as predicted protein structure may contribute more to commonly used accuracy scores than the ligand pose. Likewise, for pocket conditioned ligand generative tasks, similarity between ligand structure and pocket interactions can be expected to contribute to leakage more than receptor sequence similarity. Figure A1 illustrates that using PLINDER similarity annotations, the test and validation set can be further stratified into

task-specific subsets. These make the resource suitable for critical evaluation of various task-specific methods based on novel proteins, ligands, pockets and protein-ligand interactions. This also include the possible intersections that provide a universal testing set.

Our splitting algorithm is configurable by a set of graphs $G$, each defined on a specific similarity metric, threshold, and a neighbor depth ($D$) determining the maximum length of the shortest path between two systems to constitute leakage. The $pass\_quality()$ function decides whether a system has high enough experimental quality (see Table A2) to be considered part of the test set. The minimum neighbors $m$ helps avoid singletons or sparsely connected potentially unrealistic systems in the test set, while the maximum neighbors $M$ puts a cap on the number of systems removed from the proto-train set by one test system, to maintain an acceptable training set size. Test set redundancy is limited by allowing only $n$ number of representatives from each component cluster (for a given similarity metric and threshold) to be retained.

Apart from leakage count, we also prioritize test representatives based on the presence of systems within congeneric matched molecular series (MMS, see Appendix A.3), and the number of linked *apo* and predicted structures. For systems in test which are part of a congeneric MMS, all members of that series passing quality criteria are moved to test, and the corresponding leaked systems are removed from proto-train. The obtained proto-train set is further split into training and validation sets based on random 90/10 split of component clusters for a given metric and threshold (`pocket_shared` $\geq 50$ weak components was used for the splits presented in this manuscript). The algorithm was applied with the four configurations listed in Table A10 on all PLINDER *holo* systems. We select one of the configurations most applicable to blind-docking (configuration 1) for re-training DiffDock to study the effect of leakage on model performance.

**Fraction of leaked systems:** For a given train-test split,

| DATASET | PLINDER | PDBBIND | DOCKGEN |
|---|---|---|---|
| SYSTEMS | 449,383 | 30,337 | 41,791 |
| PDB IDS | 110,791 | 19,007 | 16,881 |
| PASS QUALITY | 113,498 | 10,818 | 19,355 |
| RECEPTORS | 74,256 | 5,425 | 7,961 |
| SMILES | 51,573 | 15,279 | 91,74 |
| CCD CODES | 46,988 | 15,064 | 9,164 |
| CATH | 1,641 | 649 | 603 |
| SCOP2B | 11,154 | 2,423 | 2,817 |
| ECOD T NAME | 1,332 | 528 | 478 |
| ECOD T ID | 4,458 | 1,444 | 1,513 |
| PROTEIN KINASE | 297 | 184 | 174 |
| KINASE INHIBITORS | 48,064 | 4,682 | 5,605 |
| APO LINKED | 98,473 | - | - |
| AFDB LINKED | 205,300 | - | - |

*Table 1.* **Protein-ligand interaction dataset comparisons.** Only *holo* systems are listed for PLINDER. See Appendix A.4 for PDBBind (v2020) and DockGen mapping. "Pass quality" = systems passing the criteria listed in Table A2. "Receptors" = unique weakly connected components of 100% `protein_fident_global`. "Kinase inhibitors" = systems where a ligand is present in the kinase inhibitors list (Kanev et al., 2020). "Apo linked" and "AFDB linked" = systems linked to one or more *apo* or AFDB structure respectively, as described in Appendix A.2.

similarity metric and threshold, we find the fraction of test systems having at least one connection (an edge with similarity $\geq$ threshold for that metric) with the train set (also applied to validation vs. test and train vs. PoseBusters).

Since DiffDock is designed for single ligand docking, we focused on *holo* PLI systems involving only a single ligand and selected one system for each unique PDB ID, CCD code combination, yielding 106,745 systems across 35,255 unique ligand SMILES, referred to as PLINDER-NR (non-redundant) for DiffDock training. Split configuration 1 (with $G$ as `pocket_lddt>50`) applied to the PLINDER-NR set is referred to as PLINDER-PL50. We also created a time-based split (PLINDER-TIME), and an ECOD topology-based split (PLINDER-ECOD), both on PLINDER-NR, to compare against PLINDER-PL50 (see Appendix D for more details). PDB IDs in PoseBusters (Buttenschoen et al., 2024) are removed from the training, validation, and test sets in all three splits to evaluate DiffDock model performance on PoseBusters. DiffDock was retrained on the three splits as described in Appendix E.

## 3. Results

### 3.1. PLINDER in numbers

At the time of writing of this article, PLINDER contains 1,344,214 PLI systems extracted from 162,978 PDB entries of which 449,383 are *holo* systems (with the remaining consisting of 573,169 systems with common experimental artifacts, 318,060 ion systems, and 3,602 systems con-

taining more than five protein and/or ligand chains, see Appendix A.1 for system classification). Within the *holo* systems, 26% have more than one ligand, 25% have more than one interacting protein chain, and 34% of systems determined by X-ray diffraction pass the X-ray high quality criteria listed in Table A2. As the curation workflow operates on the entire PDB, the collection and labelling of *holo* systems allowed us to simultaneously identify 564,240 PDB chains as being *apo*. Thus, PLINDER also provides an automatically curated dataset of *apo* chains with no detectable ligand interactions (except for artifacts or ions). Table 1 shows how PLINDER compares to the commonly used PDB-Bind (Wang et al., 2005) and DockGen datasets (Corso et al., 2024). Each system in PLINDER has over 500 annotations across the categories listed in Appendix A.

Of the 615,932 ligands in *holo* systems covering 46,988 unique CCD codes, 233,760 (37%) pass the Lipinski Ro5 criteria, 146,444 (23%) have a covalent linkage, 122,741 (19%) are cofactors, 105,836 (17%) are oligo-saccharides, -nucleotides or -peptides, and 55,987 (9%) are fragments. 15,383 systems are part of 2,117 congeneric MMS, each with at least three ligands containing a common core.

### 3.2. PLINDER splits

As shown in Table 2, the PLINDER-PL50 split exhibits the lowest leakage levels between training and test sets compared to PLINDER-TIME and PLINDER-ECOD. The `pocket_lddt` graph used for splitting eliminates edges between train and test for this metric at threshold above 50%, but also effectively removes most connections based on interactions, pocket location, and protein sequence similarity. Although similar ligands appear in both sets, they bind to different protein pockets which forms an acceptable test-case for the rigid-body docking task and method being assessed.

The PLINDER-ECOD split, despite lower leakage than PLINDER-TIME, reveals issues with incomplete domain annotations. In this case, as systems with no available ECOD annotations were chosen for test, many of these do possess the same domains and pockets seen in the training set. The analogous DockGen split demonstrates that a careful and complete assignment of ECOD domains can reduce leakage but restricts the data that can be assigned and still relies on manual curation efforts.

The large PLINDER dataset size also ensures high diversity sets irrespective of splitting strategy. All PLINDER-PL50 test systems meet high-quality criteria by design, contrasting with 21% and 19% for PLINDER-ECOD and PLINDER-TIME, ensuring reliable ground truth. Table 2 also compares the numbers of train, validation and test systems across the different splits demonstrating our capabilities of creating PLINDER-PL50 dataset-split with a high quality test set con-

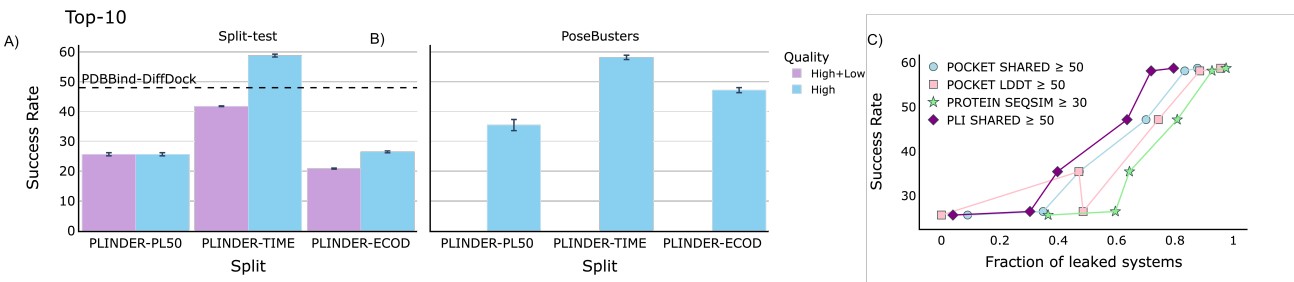

*Figure 2.* **DiffDock performance**. Success rate as the percentage of test systems with at least one pose with RMSD <2 Å from reference in the Top-10 poses when trained of different training sets and tested on: **A)** corresponding test sets, and **B)** PoseBusters set. **C)** Relationship between DiffDock success rate (high quality test sets) from A) and B) and their fraction of leaked systems as reported in Table A11.

*Table 2.* Dataset train vs. test split/PoseBusters fraction of leaked systems

| SPLIT SET | PLI SHARED $\geq 50$ | POCKET LDDT $\geq 50$ | POCKET SHARED $\geq 50$ | PROTEIN GLOBAL LDDT $\geq 50$ | PROTEIN SEQSIM $\geq 30$ | LIGAND SIMILARITY $\geq 30$ | No. TRAIN / VAL / TEST | TEST PASS QUALITY% |
|---|---|---|---|---|---|---|---|---|
| | | | | VS. TEST SET | | | | |
| PDBBIND-ORIGINAL | 0.91 | 1.00 | 1.00 | 1.00 | 1.00 | 0.62 | 22,365 / 7,549 / 423 | 50.12 |
| PDBBIND-DIFFDOCK | 0.43 | 0.76 | 0.73 | 0.76 | 0.80 | **0.43** | 25,442 / 1,570 / 236 | 22.46 |
| DOCKGEN | 0.04 | 0.08 | **0.05** | 0.08 | **0.18** | 0.64 | 40,916 / 285 / 590 | 50.00 |
| PDBBIND-LP | 0.77 | 0.87 | 0.86 | 0.89 | 0.94 | 0.40 | 18,152 / 3,906 / 7,265 | 40.37 |
| PLINDER-TIME | 0.80 | 0.96 | 0.88 | 0.95 | 0.98 | 0.54 | 76,950 / 11,392 / 11,412 | 19.28 |
| PLINDER-ECOD | 0.30 | 0.49 | 0.35 | 0.49 | 0.60 | 0.52 | 77,411 / 10,169 / 12,174 | 20.81 |
| PLINDER-PL50 | **0.04** | **0.00** | 0.09 | **0.01** | 0.37 | 0.58 | 57,602 / 3,453 / 3,729 | **100.00** |
| | | | | VS. POSEBUSTERS | | | | |
| PDBBIND-DIFFDOCK | 0.52 | 0.69 | 0.65 | 0.70 | 0.78 | **0.59** | 25,442 / 1,570 / 308 | 100.00 |
| PLINDER-TIME | 0.72 | 0.88 | 0.83 | 0.88 | 0.93 | 0.66 | 76,950 / 11,392 / 308 | 100.00 |
| PLINDER-ECOD | 0.64 | 0.74 | 0.70 | 0.75 | 0.81 | 0.65 | 77,411 / 10,169 / 308 | 100.00 |
| PLINDER-PL50 | **0.40** | **0.47** | **0.47** | **0.48** | **0.64** | 0.64 | 57,602 / 3,453 / 308 | 100.00 |

Note that the 41,961 PLINDER-NR systems are removed from the PLINDER-PL50 split by the splitting algorithm to avoid leakage.

taining ten times more complexes than the commonly used PoseBusters set while still maintaining low leakage to the corresponding training set.

### 3.3. DiffDock performance on different splits

DiffDock was re-trained using NVIDIA BioNeMo (bio) on PLINDER-PL50, PLINDER-ECOD and PLINDER-TIME splits as described in Appendix E. Its performance was evaluated against corresponding test sets, and the standardized PoseBusters benchmark set. All evaluations utilize single A100 GPU inference configurations. Inference protocols involve generating poses of the ligand on a protein and attempting to identify the protein pocket and corresponding ligand configuration from the ground truth test set. Poses are ranked by a confidence model pre-trained as previously described by Corso et al. (2023), and an RMSD is calculated for each pose. Because we trained new score models but used a pre-existing confidence model, the Top-1 pose selection may be erroneous. Therefore, we generate 10 poses, and report Top-1 and Top-10 poses that have RMSD <2Å. The Top-1 pose will be a selection from the unoptimized confidence model, while the Top-10 pose considers the aggregate likelihood of all generated poses, thereby mitigating

the influence of the confidence model. This discrepancy in reported success underscores the necessity of optimizing the confidence model for accurate reporting.

In Figure 2 and Table A12, we present the reported accuracy of our newly trained DiffDock models defined by the training sets in PLINDER-TIME, PLINDER-ECOD, and PLINDER-PL50. On the PoseBusters benchmark set, the baseline Top-1 performance reported for DiffDock trained on the PDBBind-DiffDock split is 38% (Corso et al., 2024). Simply increasing the volume of data without modifying the architecture or considering leakage boosts performance to 47.8% and 58.2% for Top-1 and Top-10 respectively, highlighting the critical role of training set size and diversity in the accuracy of deep learning models. However, with more principled splitting strategies in PLINDER-PL50 and PLINDER-ECOD in which the test set has less information leakage to the training set, we observe a corresponding decrease in accuracy to the 15-18% range (21-26% for Top-10).

We posited that test set quality significantly impacts measured performance. As shown in Figure 2A and Appendix Table A12, DiffDock's performance drops from 45.2% to 29.2% and from 19% to 14.7% for PLINDER-Time and PLINDER-ECOD respectively, when lower quality systems are

added to the test sets. Figure 2C reveals a linear relationship between the fraction of leaked test systems for different metrics and DiffDock performance: for most metrics higher leakage correlates with inflated apparent accuracy (success rate for Top-10 poses against high-quality test). We explore this relationship in more detail in Appendix F. Our quantifiable evidence validates the longstanding intuition regarding the importance of different similarity metrics in de-leaking.

## 4. Dataset and code availability and updates

Our automated curation workflow enables reproducible periodic updates. For each system, apart from annotations, we provide incremental additions to the protein similarity and ligand similarity datasets for comparative queries, and assign clusters directly based on the new similarity data. On a semi-annual basis, new systems will be consolidated into the existing similarity datasets and clustering will be re-run from scratch and released as a new version. The dataset itself is made available to the public with a CC-BY 4.0 license and hosted on a Google Cloud Storage bucket. The schema of the available dataset will be updated as changes are made and a document describing the draft schema is available here.

The PLINDER source code is Apache 2.0 licensed and can be downloaded at https://github.com/aivant/plinder. Appendix C describes the engineering and software choices which allowed fast and efficient dataset curation and graph querying. Apart from dataset curation, we provide software to evaluate predictions of protein-ligand complexes of any size against reference PLINDER systems using a range of accuracy metrics for ligand (RMSD, lDDT-PLI, PoseBusters), pocket (lDDT-LP), and protein (lDDT, oligomeric scores) levels (Biasini et al., 2013; Studer et al., 2023; Robin et al., 2023; Buttenschoen et al., 2024). In the near future, we plan to release software supporting efficient data loading and diversity sampling of PLINDER, and the PLINDER-2024 split with stratified validation and test sets to make PLINDER readily accessible for the machine learning community.

## 5. Current limitations and future directions

As the global scientific community continues to generate new experimental data, this work, too, reports an on-going effort to provide the most comprehensive resource for methods in the field of protein-ligand interaction prediction.

We are working towards additional data annotations, such as classical docking scores, cross-docking scores, measured and predicted binding affinities, cryptic pocket and promiscuous ligand labels, to prioritize and stratify test sets with varying properties and auxiliary task labels. We are exploring data augmentation strategies to increase the diversity of interactions, pockets, and folds covered using experi-

mental structures without ligands (Corso et al., 2024) and prediction methods (Voitsitskyi et al., 2024), for which our interaction and leakage detection algorithms are already applicable. While the current release links only a single AlphaFold model to each system, we will make use of the redundancy within UniProt to link multiple predicted structures from the AlphaFold Database to each system. As we observed cases where PLIP failed to detect high similarity between near-identical pockets, we will also explore alternative protein-ligand interaction profilers for curating PLI systems. Our focus on filtering high-quality test systems favours smaller molecules and may underrepresent protein or ligand classes for which only low quality structures are available. Ideally, atom-level weighting would be used in accuracy metrics instead, to be explored in future projects. In addition, while we only utilize quality information for X-ray structures, validation reports for electron microscopy structures now include per-residue Q-scores (Pintilie et al., 2020), which could be incorporated into similar criteria.

To enable a fair assessment of methodological advances we plan to add a leaderboard for commonly adopted methods trained and tested on PLINDER datasets. This will include models trained on the full PLINDER data split, including diversity sampled redundant and augmented systems beyond the subsets described for DiffDock training in this manuscript. We also aim to cover more realistic assessment scenarios of cross-docking or predicting poses within *apo* and predicted receptor structures, with predicted pockets stratified by the extent of conformational change.

## 6. Conclusion

We present PLINDER a large, comprehensive and automated dataset resource for protein-ligand interactions. We demonstrate the value in scalable similarity measures between protein-ligand complexes and a splitting algorithm that prioritises test set quality and low information leakage. By retraining DiffDock on various splits, we show that our methods and results provide a solid foundation for dataset generation, as well as measuring and addressing both quality and dataset leakage in a quantitative and tunable manner.

## 7. Acknowledgements

This work has received funding from the LIGATE public-private consortium supported by the European High-Performance Computing Joint Undertaking (JU) under grant agreement No 956137, SIB Swiss Institute of Bioinformatics (https://www.sib.swiss/) and the Biozentrum of the University of Basel (https://www.biozentrum.unibas.ch/). The JU receives support from the European Union's Horizon 2020 research and innovation programme and Italy, Sweden, Austria, the Czech Republic, Switzerland.

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

# A. Curation and Annotation

## A.1. Ligand and *holo* system annotation

In order to characterize our systems, we identified small-molecules that are not considered PLINDER ligands - artifacts and single (non-hydrogen) atom or ion entities. All entities containing a single atom, excluding basic organic elements C, H, N, O, P, S, were kept as auxiliary entity, which we refer as an "ion", irrespective of its oxidation state (eg. $Fe^{2+}$, Xe, $Cl^-$). These entities were considered as part of the pocket, if they were within 4 $\mathring{A}$ distance from the ligand, but not considered as a ligand (hereafter referred to as "ions"). To enrich for biologically and therapeutically relevant ligands, we further excluded molecules that contain few atoms (less than 2 carbons or less than 5 non-hydrogen atoms), are highly charged (absolute charge more than 2), have long unbranched hydrocarbon linkers (longer than 12), contain unspecified atoms or are common experimental buffer artifacts. The artifact entries were curated based on commonly used artifact definitions (Zhang et al., 2024) and including additional commonly occurring buffer reagents, but excluding biologically relevant co-factors or sugars (eg. NAG). The list used is provided in Table A4. The ligand inclusion criteria is summarized in Table A3, if the molecule does not pass the criteria - it is labelled as an "artifact". All PLI systems which have at least one ligand that passes the curation criteria form our dataset of *holo* systems.

Each detected ligand was assigned a unique canonical SMILES. For single residue ligands we used the Chemical Component Dictionary (CCD) (Westbrook et al., 2015) to identify the SMILES. If the ligand name was not present in the CCD or it consisted of multiple residues, we used the resolved SMILES value from PLIP that was further standardized via RDKit canonicalization. We then used the canonical SMILES to calculate ligand descriptors for Lipinski criteria and other commonly used metrics such as the quantitative estimate of drug-likeness (QED) (Bickerton et al., 2012). Finally, the SMILES were used to assign bond orders to resolved ligand atoms. When some atoms were missing in the resolved structure, we used substructure matching to identify the relevant molecular fragments to assign the bond orders. In case of failure to assign bond orders from SMILES, we used OpenBabel (O'Boyle et al., 2008) to assign the bond orders. Despite our best efforts not all ligands were successfully processed by RDKit. In Table A5 we report the numbers for PLINDER total and RDKit-processable ligand SMILES. Note that these numbers also include ions and artifacts not counted in Table 1.

Table A6 provides and overview for numbers of protein and ligand chains per *holo* system in PLINDER dataset.

## A.2. *Apo*, predicted and cross-docking structures

Each system is associated where possible to related (1) *apo* chains, (2) predicted structures from the AlphaFold database (Varadi et al., 2022; Jumper et al., 2021) and (3) other *holo* systems.

*Apo* protein chains within each PDB entry are defined as those with a different entity identifier to any protein chain in that entry participating in a *holo* system. These are collected across the entire PDB to form the database of *apo* chains. Similarly, a database of predicted structures is constructed using the sequences and AFDB models of all UniProt IDs associated with PDB structures. A similar Foldseek and MMseqs search as described in Section 2.2 is performed for the *holo* systems against these two databases with min_seq_id set to 0.9 and minimum query coverage set to 0.9, to calculate all the protein scores, as well as the pocket_lddt and pocket_identity scores (Note that the remaining scores cannot be calculated as they depend on the presence of a ligand and binding pocket in the target). Cross-docking structures are instead obtained for each system using the similarities already calculated in 2.2. The *apo*, predicted and cross-docking hits are further restricted with a filter of 95% pocket_identity.

Each linked structure is superposed to the corresponding system and additional metrics are calculated based on transplanting the ligand from the system to the linked structure. These include the RMSD of the superposition as well as the lDDT-LP, lDDT-PLI (Robin et al., 2023) and PoseBusters checks (Buttenschoen et al., 2024) of the transplanted ligand.

## A.3. Matched molecular series

To enable evaluation of lead optimization use-cases, we detected PLI systems containing matched molecular series (MMS), where the ligands in the set all contain the same constant core and each pair within the set only differ by a single chemical transformation. This allows evaluating the performance in use-cases, where both pockets and some members of a chemical series are known and methods are used to prioritize changes to this series that optimize binding. This is a key part of typical lead optimization stages of drug development, and this annotation allows for fair evaluation of methods that make use of this prior knowledge available at inference time.

Using all the annotated ligands, we generated the mm-pdb index using default parameters. This index maps ligands that are matched molecular pairs to their shared core scaffold. We dropped scaffolds which were fragmented or contained less than 5 atoms. Systems with pockets in the same 95% protein_identity and 100% pocket_identity_shared strong components con-

| Category | Annotations |
|---|---|
| Identifiers | PDB ID, biounit, interacting protein chains and residue numbers, ligand chains |
| Entry information | Release date, oligomeric state, determination method, keywords, pH |
| System information | System type (*holo*, artifact, ion, see Appendix A.1), CCD code, unique CCD code (de-duplicated for identical molecules with different CCD codes), canonical SMILES, resolved SMILES |
| Ligand properties | Molecular weight, Wildman-Crippen ClogP (Wildman & Crippen, 1999), hydrogen bond donor and acceptor counts, number of rings, number of heavy atoms, QED (Bickerton et al., 2012), covalent bonds, passing Lipinski's rule of five (Ro5) criteria, detected as fragment, detected as oligo-peptide, -saccharide or -nucleotide, presence in cofactor list, presence in artifact list, presence in list of kinase inhibitors (Kanev et al., 2020), BIRD ID (Dutta et al., 2014), detected PLIP interactions (Adasme et al., 2021), PoseBusters checks (Buttenschoen et al., 2024) |
| Protein properties | CATH (Orengo et al., 1997), ECOD (Cheng et al., 2014), SCOP (Murzin et al., 1995), Pfam (Mistry et al., 2021), UniProt (Consortium, 2019), Kinase (Kanev et al., 2020), PANTHER (Thomas et al., 2022) for each protein chain in the entry and also aggregated to the system pocket |
| Entry quality | Resolution, $R$, $R_{free}$, clash score, % Ramachandran outliers, % rotamer outliers, % RSRZ outliers, data completenessMolProbity score (Chen et al., 2010) |
| Per-residue quality | RSR, RSCC, RSRZ,occupancy, missing/resolved atom counts, outliers (clash, geometry, density, chirality) and alternative configuration count. Per-residue quality metrics are also aggregated across ligands, pockets, and chains. |
| Similarity clusters | Protein, pocket, PLI, and ligand-level components at thresholds of 50%, 70%, 95% and 100% for metrics listed in Appendix B |
| Matched molecular series | Congeneric series IDs, core scaffolds and transformations, see Appendix A.3 |
| Linked structures (*apo*, AFDB) | Associated similarity scores, CIF files superposed to each *holo* system, superposition RMSD, lDDT-LP, lDDT-PLI (Robin et al., 2023), and PoseBusters results based on transplanted ligand, see Appendix A.2 |
| System files | MMCIF and PDB files of all ligand chains, interacting protein chains and interacting waters extracted from the biounit. SDF files of each ligand chain. Note: chains are renamed for PDB format. |

*Table A1.* **Annotations available per system**. Annotation availability is indicated with a copper color scheme, where light orange is zero availability, **brown** is 50% availability and **black** is available in all systems. For boolean annotations such as "presence in cofactor list", the color indicates how many systems have the annotation as True.

*Table A2.* X-ray high quality criteria for PLI systems, based on a combination of relevant previously defined criteria (Warren et al., 2012; Leemann et al., 2023; Buttenschoen et al., 2024).

| PROPERTY | TEST VALUE |
|---|---|
| **ENTRY** | |
| RESOLUTION | $\leq 3.5$ |
| $R$-FACTOR | $\leq 0.4$ |
| $R_{free}$ | $\leq 0.45$ |
| $R - R_{free}$ | $\leq 0.05$ |
| **LIGAND AND POCKET** | |
| NO UNRESOLVED HEAVY ATOMS | TRUE |
| NO ALTERNATIVE CONFIGURATIONS | TRUE |
| AVERAGE OCCUPANCY | $\geq 0.8$ |
| AVERAGE RSCC | $\geq 0.8$ |
| AVERAGE RSR | $\leq 0.3$ |
| **LIGAND** | |
| NO CLASH OUTLIERS | TRUE |

*Table A3.* Non-artifact ligand classification criteria

| PROPERTY | VALUE |
|---|---|
| IS A SINGLE ATOM (ION) | FALSE |
| NON-H ATOM COUNT | $> 5$ |
| C ATOM COUNT | $> 2$ |
| ABSOLUTE CHARGE | $\leq 2$ |
| UNBRANCHED HYDROCARBON LINKER LENGTH | $\leq 12$ |
| UNSPECIFIED ATOM (*) COUNT | 0 |
| CCD CODE (OR SYNONYM) IN TABLE A4 | FALSE |

### A.4. Comparison with other PLI datasets

We selected PDBBind (Wang et al., 2005), and Dock-Gen (Corso et al., 2024) to compare PLINDER against PLI datasets that have previously been used for training and evaluation of deep learning methods. These datasets are mapped to PLINDER using a combination of PDB IDs and CCD codes, with the first biounit containing the ligand considered for PDBBind and specified biounit identifiers for DockGen.

taining ligands sharing the same scaffold are collected into congeneric MMS.

*Table A4.* CCD codes of ligands that are treated as artifacts

02U, 12P, 13P, 144, 15P, 16P, 1EM, 1PE, 1PG, 1PS, 2DP, 2JC, 2NV, 2OP, 2PE, 32M, 33O, 3HR, 3PG, 3SY, 3V3, 543, 6JZ, 6PE, 7E8, 7E9, 7I7, 7N5, 7PE, 7PG, 7PH, 90A, 9FO, 9JE, 9YU, AAE, ABA, AE3, AE4, AGA, AKR, AUC, B3H, B3P, B4T, B4X, BAM, BCN, BDN, BE7, BEN, BET, BEZ, BGL, BHG, BNG, BNZ, BOG, BTB, BU1, BXC, C10, C14, C8E, CAC, CAD, CAQ, CD4, CE1, CE9, CHT, CIT, CN3, CN6, CPS, CXE, CXS, D10, D12, D1D, D22, DAO, DD9, DDQ, DDR, DEP, DET, DHB, DHJ, DIO, DKA, DMF, DMI, DMR, DOX, DPG, DR6, DRE, DTD, DTT, DTU, DTV, E4N, EAP, EEE, EPE, ETE, ETF, ETX, F09, F4R, FJO, FTT, FW5, GLV, GOL, GVT, GYF, HAE, HAI, HCA, HCS, HED, HEX, HEZ, HP6, HSG, HSH, HT3, HTG, HTH, HTO, HZA, I3C, ICT, IHP, IHS, IMD, IPH, JDJ, K12, KDO, L1P, L2C, L2P, L3P, L4P, LAC, LDA, LI1, LMR, LMT, LMU, LUT, M2M, MAC, MAE, MB3, MBN, MBO, MC3, ME2, MEG, MES, MLA, MLI, MLT, MPD, MPO, MRD, MSE, MYR, N8E, NBN, NET, NEX, NHE, O4B, OCT, OES, OGA, OP2, OTE, OXM, P03, P15, P1O, P22, P25, P2K, P33, P3G, P4C, P4G, P4K, P6G, PA8, PC8, PD7, PE3, PE4, PE5, PE6, PE7, PE8, PEG, PEP, PEU, PEX, PG0, PG4, PG5, PG6, PG8, PGE, PGF, PGO, PGR, PHB, PHQ, PL9, PLC, PMS, PPI, PQ9, PQE, PTD, PUT, PVO, PX2, PX4, QGT, QJE, QLB, RG1, RWB, SAR, SEP, SGM, SIN, SOG, SP5, SPD, SPJ, SPM, SPZ, SQU, SRT, TAM, TAR, TAU, TBU, TCE, TCN, TEA, TFA, THE, TLA, TMA, TOE, TPO, TRD, TRS, UMQ, UND, V1J, VX, XAT, XP4, XPA, XPE, Y69.

*Table A5.* Ligand annotation overview in PLINDER

| Ligand Type | Number of Unique Items |
| --- | --- |
| Ligand SMILES | 54,089 |
| Ligand RDKit Canonical SMILES | 53,543 |
| Ligand CCD Code | 48,271 |

*Table A6.* Number of protein chains and ligand chains in PLINDER *holo* systems

| PROTEIN CHAINS | LIGANDS | SYSTEM IDS |
| --- | --- | --- |
| 1 | 1 | 257836 |
|   | 2 | 57312 |
|   | 3 | 16854 |
|   | 4 | 4508 |
|   | 5 | 1417 |
| 2 | 1 | 62977 |
|   | 2 | 25852 |
|   | 3 | 5985 |
|   | 4 | 2691 |
|   | 5 | 809 |
| 3 | 1 | 7129 |
|   | 2 | 2057 |
|   | 3 | 1163 |
|   | 4 | 312 |
|   | 5 | 83 |
| 4 | 1 | 1347 |
|   | 2 | 481 |
|   | 3 | 88 |
|   | 4 | 126 |
|   | 5 | 29 |
| 5 | 1 | 201 |
|   | 2 | 23 |
|   | 3 | 6 |
|   | 4 | 19 |
|   | 5 | 78 |

Mapping details are shown in Table A7 and Table A8. Systems with annotations in PLINDER that differ from PDBBind or Dockgen are referred to as inconsistent with PLINDER. For example, some ligands labeled as peptides with four amino acids in PDBBind were found to have five or six amino acids in PLINDER, or were identified with entirely different ligand CCD codes. PDB IDs that did not pass the PLINDER processing pipeline are referred to as not in PLINDER. These consist of cases in PDBBind and DockGen containing peptides longer than 11 amino acids, which are not considered as ligands in PLINDER.

## B. Similarity measures

### B.1. Protein similarity

Foldseek and MMSeqs searches were run on all PDB chains present in all systems (E-value < 0.01, min_seq_id 0.2, max_seqs 5000). The fraction of identical residues and query coverage were saved along with the full query and target alignments. Additionally for Foldseek, alignment lDDT and lDDT scores per aligned residue were saved. These search results are used to define the following protein level similarities for a given alignment, with the highest value being taken for each if both Foldseek and MMSeqs alignments

are found:

- **protein_lddt**: lDDT score

- **protein_identity**: Percentage of identical residues

- **protein_seqsim**: Sequence similarity

- **protein_qcov**: Query coverage

- **protein_lddt_global, protein_identity_global, protein_seqsim_global**: Corresponding similarities multiplied by the query coverage, approximates the value for the global alignment.

For systems with more than one interacting protein chain, greedy chain mapping is performed using the `protein_lddt_global` score where available and `protein_identity_global` otherwise.

The protein similarity score ($S_{a,b}$) between systems $a$ and $b$ is then calculated as a weighted mean across the mapped pairs of chains $(i, j)$, with the weight ($l_i$) defined by the

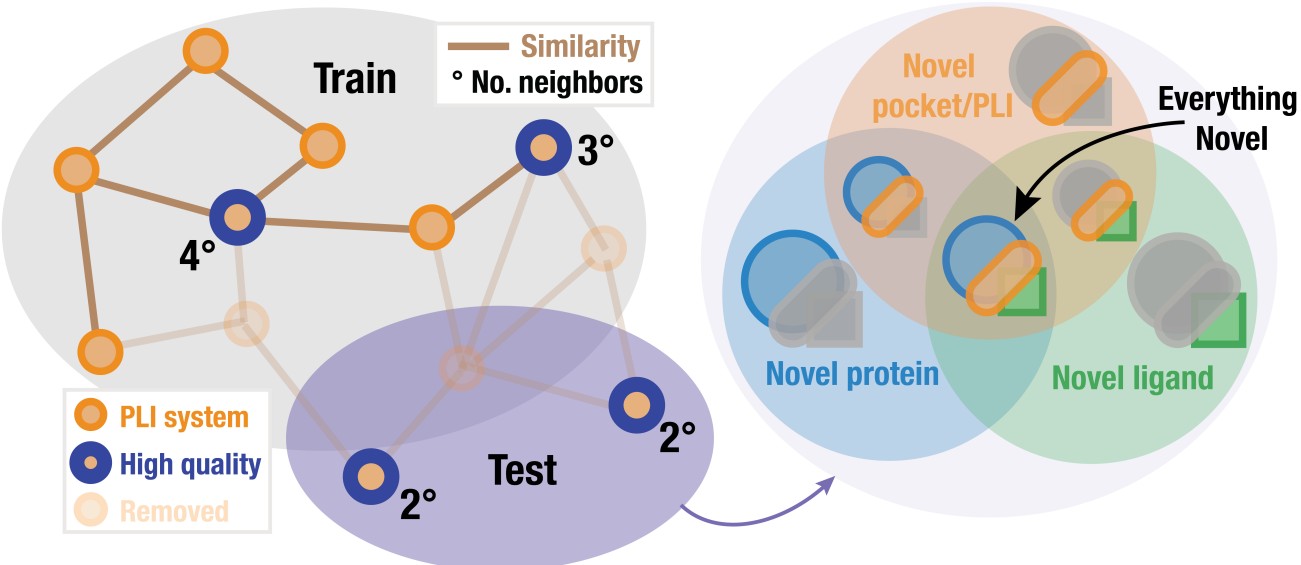

*Figure A1.* PLINDER test sets can be further stratified into subsets using extensive system similarity annotations. The test and validation sets can be further subsplit into subsets of novel protein, ligand and protein-ligand interaction systems, including their intersections for potential multi-task testing.

*Table A7.* PDBBind mapping to PLINDER details

| PDBBind 2020 | |
| --- | --- |
| total PDB ID + ligand CCD combinations | 19443 |
| total PDB IDs | 19443 |
| total ligand CCDs | 12930 |
| PDB IDs in PLINDER | 19007 |
| PDB IDs + Ligand CCD ID inconsistent with PLINDER | 1375 |
| *holo* systems in PLINDER | 28184 |
| ion systems in PLINDER | 711 |
| artifact systems in PLINDER | 1442 |
| PDB IDs not in PLINDER | 436 |
| systems in PLINDER | 30337 |

*Table A8.* DockGen mapping to PLINDER details

| DockGen | |
| --- | --- |
| total PDB ID + Biounit ID + Ligand CCD combination | 26007 |
| total PDB IDs | 18504 |
| total Ligand CCD | 9338 |
| PDB IDs in PLINDER | 18391 |
| PDB IDs + Ligand CCD ID inconsistent with PLINDER | 1510 |
| *holo* systems in PLINDER | 37656 |
| ion systems in PLINDER | 3212 |
| artifact systems in PLINDER | 923 |
| PDB IDs not in PLINDER | 113 |
| systems in PLINDER | 41791 |

length of each query protein chain.

$$S_{a,b} = \frac{\Sigma_{i \in a}(l_i . S_{i,j})}{\Sigma_{i \in a} l_i}$$

### B.2. Pocket similarity

The alignment results and chain mappings obtained from protein similarity calculation are combined with information about the binding pocket, i.e residues in proximity to the ligand, to define the following pocket level scores:

- **pocket_shared** The percentage of shared and aligned binding pocket residues

$$\frac{|B_a \cap \text{Aligned}_{a \to b}(B_b)|}{|B_a|}$$

where the binding pocket residues of the query system $a$ ($B_a$) are only considered if the corresponding aligned residue in the target system $b$, $\text{Aligned}_{a \to b}$, is also in $B_b$.

- **pocket_identity** The percentage of identical residues in the query binding pocket

$$\frac{\Sigma_{n \in B_a} B_a{}^n = \text{Aligned}_{a \to b}^n}{|B_a|}$$

- **pocket_identity_shared** The percentage of shared, aligned and identical binding pocket residues

$$\frac{\Sigma_{n \in (B_a \cap \text{Aligned}_{a \to b}(B_b))} B_a{}^n = \text{Aligned}_{a \to b}^n}{|B_a|}$$

- **pocket_lddt** The mean lDDT of query binding pocket residues

$$\frac{\Sigma_{n \in B_a}\text{lDDT}_n}{|B_a|}$$

- **pocket_lddt_shared** The mean lDDT of shared and aligned binding pocket residues

$$\frac{\Sigma_{n \in (B_a \cap \text{Aligned}_{a \to b}(B_b))}\text{lDDT}_n}{|B_a|}$$

### B.3. Protein-ligand interaction similarity

We convert each PLIP interaction into a unique string, defined by the interaction type and type-specific attributes as defined in Table A9.

*Table A9.* Attributes used to define protein-ligand interaction types

| INTERACTION TYPE | ATTRIBUTES |
|---|---|
| HYDROGEN BONDS | IS PROTEIN THE DONOR?, IS THE BOND WITH A SIDECHAIN? |
| HYDROPHOBIC | - |
| SALT BRIDGES | DOES THE PROTEIN CARRY THE POSITIVE CHARGE? |
| WATER BRIDGES | IS PROTEIN THE DONOR? |
| $\pi$ STACKS | STACKING TYPE |
| $\pi$-CATION | DOES THE PROTEIN PROVIDE THE CHARGE? |
| HALOGEN BONDS | IS THE BOND WITH A SIDECHAIN? |
| METAL COMPLEXES | METAL TYPE, TARGET TYPE, COORDINATION, GEOMETRY, LOCATION |

The collection of hashed PLIP interactions is used to define a weighted Jaccard index:

$$J_w(I_a, I_b) = \frac{\sum \min(a_i, b_i)}{\sum \max(a_i, b_i)}$$

Where $I_a$ and $I_b$ are the multisets of interactions, $a_i$ and $b_i$ are the counts of each unique interaction in $I_a$ and $I_b$ respectively, $\min(a_i, b_i)$ is the minimum count of each interaction found in both sets, and $\max(a_i, b_i)$ is the maximum count of each interaction across both sets.

The **pli_shared** between two systems $a$ and $b$ is then defined by comparing the interactions at each aligning pocket residue:

$$\frac{\Sigma_{n \in (B_a \cap \text{Aligned}_{a \to b}(B_b))} J_w(I_a^n, I_b^n)}{|I_a|}$$

### B.4. Ligand similarity

To detect ligand similarity, we generate ECFP4 fingerprints with length of bits 1024 using RDKit (rdk) on ligand SMILES. We then calculated Tanimoto coefficient for each ligand pair. When more than one ligand present in a PLI system, we used the maximum similarity for each individual pair of systems. To be consistent with protein similarity

scores, as well as for storage and querying efficiency, we first ranked all the Tanimoto coefficients and saved only the top 5000 pairs. For potential leakage detection we used the 30% and 50% cutoffs previously reported for ligand based activity enrichment (Jasial et al., 2016).

## C. Dataset construction and engineering considerations

The data ingestion and data processing required to produce the complete PLINDER dataset involves large scale orchestration of a non-trivial amount of computational resources. We leverage the metaflow (met) framework to define a directed acyclic graph (DAG) of the work to be completed. This enables us to implement the processing logic in normal python code, and then simply write a metaflow.FlowSpec to distribute the work in a kubernetes (kub) cluster. We promote the FlowSpec to an argo (Arg) workflow template in order to execute the entire end-to-end pipeline in the cloud.

The RCSB rsync API makes it convenient to distribute the initial dataset ingestion along the middle two character codes of PDB IDs. However, the number of entries within a two character code are not evenly distributed, leading to uneven run times, making it a poor choice for distributing other stages in the workflow. For stages such as annotation generation and scoring, we batch chunks of PDB IDs and distribute along these chunks instead. Using either two character code or PDB ID chunks with tunable chunk sizes allows us to choose between simple distribution logic and desired total run time.

Careful consideration of distributing the workload affords the ability to run a majority of the end-to-end pipeline on commodity machines, with few exceptions. The first is for the large foldseek and mmseqs database creation steps, which are vertically scaled on a 96 core machine. Additionally, the cluster generation and splitting steps use up to 100 GB of memory because they require reading a significant portion of the protein similarity dataset into memory in order to construct the graphs.

Downstream stages of the workflow that leverage the similarity and cluster datasets initially suffered from a substantial bottleneck simply reading the data from disk. Converting the data to parquet files and partitioning the data based on commonly applied query filters reduced query times from 30 minutes or more to under 10 minutes. Subsequent adoption of the duckdb(Raasveldt & Muehleisen) embedded database engine with its query optimizer further reduces query times by a factor of up to two to four. Currently, materialization of the data into a data frame is the slowest step for large queries (100 million to 1 billion records). Finally, since duckdb supports standard query language (SQL), it

is possible to push complex aggregations that are currently done in python into the query itself, including cross-dataset joins, further reducing the cost in both time and memory of querying these data.

## D. Comparing splitting methods

To evaluate generalizability of deep learning model performance, along with PLINDER the dataset itself, we propose a splitting algorithm 1 which can separate dissimilar protein-ligand interactions, protein and ligands systems into training, validation, test sets while maintaining test set quality and train and test set diversity. We applied this algorithm with four different configurations, listed in Table A10 on the entire PLINDER dataset. For all splits, $m = 2$, $M = 400$, $C$ is defined by the strongly connected components of `pli_shared` $\geq 70$, and $n = 5$. The resulting proto-train sets were divided into train and validation sets based on weakly connected components of `pocket_shared` $\geq 50$. As shown by the fraction of leaked systems for different metrics and thresholds in Table A10, splitting by different similarity graphs removes leakage for that metric and to varying extents also removes connections based on other metrics. For example, splitting based on the `pocket_shared` $\geq 20$ graph (configuration 2) only ensures that train and test systems do have a ligand-binding pocket in the same location. As seen by the resulting fraction of leaked systems for `protein_seqsim` for this configuration, this implies that similar proteins are in train and test (but necessarily with ligands bound in different locations). Similarly, as none of these configurations performed splitting using the ligand similarity graph, similar ligands are in both train and test for all splits. Future efforts will go towards optimizing thresholds and the $m$ and $M$ parameters to obtain a required test-set size, combining different component clusters to further separate train and validation sets, and combining different graphs to obtain stratified test sets for task-specific assessment.

The first configuration, which distinguishes similar pockets even for cases where ligands bind to alternative sites, was applied to a non-redundant single-ligand subset (PLINDER-NR, described in Section 3.2) to create the PLINDER-PL50 split which was used for retraining Diffdock. To better demonstrate the advantage of this split method, we evaluate our split against two more splits on PLINDER-NR also used to retrain DiffDock, four previously detailed splits mapped to PLINDER as described in Appendix A.4, and one additional high quality test set:

1. **PLINDER-Time**: Using January 1, 2021, as the cutoff date for the training set, all PLINDER-NR systems submitted after this date up to 2022-04-19 form the validation set, and those submitted afterward up to 2022-04-19 form the test set.

2. **PLINDER-ECOD**: This split was generated using the pocket-level ECOD topology (t-name) annotations. Systems from PLINDER-NR were first grouped by ECOD t-name and sorted by its number of members, the biggest group was added to train until the training set had 80% of systems. The rest of the groups were divided evenly into validation and test sets, along with systems with unknown t-name.

3. **DockGen split**: Another ECOD t-name derived split on a different dataset provided by DockGen (Corso et al., 2024).

4. **PDBBind-Original split**: This split uses the "general", "refined" and "core" sets of PDBBind (Liu et al., 2015) as train, validation and test sets respectively.

5. **PDBBind-DiffDock split**: This split was generated by removing ligands and receptors similar to the training set from the time-split PDBBind test set, used in DiffDock (Corso et al., 2023).

6. **PDBBind-LP**: A cleaned and reorganzied PDBBind split of non-covalent binders controlling for leakage defined by high protein sequence and ligand similarity (Li et al., 2024).

7. **PoseBusters** has recently gained popularity as an additional benchmark set for evaluating docking methods, due to its focus on higher quality systems. To include evaluations against PoseBusters, we excluded all PoseBusters PDB IDs from PLINDER-ECOD and PLINDER-TIME splits.

In Table A11 we present the fraction of leaked systems (defined in Section 2.3) for each split for different metrics and thresholds. We note that RDKit failed to obtain ECFP4 for a number of SMILES, which might effect the reported ligand similarity fraction of leaked systems values. We report these numbers for the different PLINDER splits in Table A14.

## E. Retraining DiffDock

We re-train DiffDock using the NVIDIA BioNeMo Framework (bio) (FW) with an identical model size of 20.2M parameters for original DiffDock comparison. A Fused Adam optimizer was used a with learning rate of 0.001, and decay rates ($\beta_1$ and $\beta_2$) 0.9 and 0.999. A new feature of DiffDock in the BioNeMo FW is an Adaptive Batch Sampler during training. Due to the variable size of protein-ligand complexes, the memory requirements when loading the heterograph can vary. This algorithm pre-computes the memory requirement of each complex, and shuffles the batches to

accommodate the memory overhead. Effective batch size is 12-13. All models were trained on eight 80GB A100 GPUs for 200-400 epochs, and fully converge in 24-30 hours. For a full list of model hyperparameters, see Table A13.

## F. Performance vs Leakage analysis

Previous studies suggested that reported deep learning method performance in general ligand pose prediction is overestimated due to information leakage between the train and evaluation sets (Corso et al., 2024; Li et al., 2024). This information leakage is commonly detected by employing similarity metrics that are used to compare test systems to those encountered in training. We employed multiple metrics for this purpose (see Appendix B) to estimate the extent of leakage for PLINDER and other dataset splits.

To assess the generalization ability, several recent studies used high performing examples with quoted ligand or protein sequence dissimilarity above or below a certain cutoff. For example, Krishna et al. (2024) used below 30% protein sequence identity and below 0.5 ligand Tanimoto similarity, to consider dissimilar protein and ligand, respectively. However, it remains unclear to what extent different similarity metrics inflate the performance.

As we chose to remove the high quality PoseBusters from PLINDER train / validation / test splitting, none of our PLINDER splitting approaches (time, ECOD, PL50) have dealt with active de-leaking for this subset. As a result, this provides a good representative sample for leakage-performance assessment using three different DiffDock models trained on three PLINDER splits. For each model we estimate the success rate comparing the most accurate (lowest RMSD to reference) prediction in Top 10 poses to 2 Å threshold. Figure A2 demonstrates the relationship across different metrics and thresholds relative to reported performance. In Figure A2A, we show the mean excess success rate observed for "leaked" test systems relative to the baseline (all test systems) as a function of "leakage", determined by train-test distance cutoff. Likewise, in Figure A2B and Figure A2C, we show success rate enrichment for leaked systems compared to all test systems and change in best pose RMSD as a function of leakage cutoff for various similarity metrics. Finally, Figure A2D shows the cumulative leakage fraction as a result of distance to train set. This analysis reveals that DiffDock performance has varied sensitivity for different type of information leakage. The pocket and PLI similarity metrics show the most sensitive difference in performance. Systems that share higher than 50 PLI similarity seem to contribute the most significantly to overestimated performance, while ligand similarity is hardly affecting DiffDock success. It is somewhat intuitive that for a rigid docking task the pocket specific interactions form the most sensitive information that result in inflated performance if leaked. However,

for different model architectures and especially for different tasks - this assessment should be repeated independently. To enable such evaluation and provide model developers with a better understanding of the factors affecting their prediction methods, we provide scripts to generate such leakage vs. performance plots across different similarity metrics and performance measures.

*Table A10.* Split configurations used and resulting leakage metrics

| PARAMETER | CONFIGURATIONS | | | |
|---|---|---|---|---|
| | 1 | 2 | 3 | 4 |
| $G$ | [POCKET LDDT $\geq$ 50] | [POCKET SHARED $\geq$ 20] | [PROTEIN SEQSIM $\geq$ 30] | [PLI SHARED $\geq$ 20, POCKET SHARED $\geq$ 50, PROTEIN LDDT $\geq$ 70] |
| $D$ | [2] | [2] | [2] | [2, 2, 1] |
| SPLIT | NUMBER OF SYSTEMS | | | |
| TRAIN | 279297 | 248849 | 339791 | 255463 |
| TEST | 14491 | 16910 | 4932 | 15132 |
| VAL | 19452 | 10131 | 17666 | 13896 |
| REMOVED | 122384 | 159734 | 73235 | 151133 |
| SPLIT-PAIR | FRACTION OF LEAKED SYSTEMS | | | |
| **PLI SHARED $\geq$ 50** | | | | |
| TRAIN VS. TEST | 0.03 | 0.00 | 0.00 | 0.00 |
| VAL VS. TEST | 0.00 | 0.00 | 0.00 | 0.00 |
| TRAIN VS. VAL | 0.06 | 0.02 | 0.02 | 0.03 |
| TRAIN VS. POSEBUSTERS | 0.52 | 0.39 | 0.77 | 0.46 |
| **POCKET LDDT $\geq$ 50** | | | | |
| TRAIN VS. TEST | 0.00 | 0.58 | 0.00 | 0.14 |
| VAL VS. TEST | 0.00 | 0.15 | 0.00 | 0.03 |
| TRAIN VS. VAL | 0.89 | 0.77 | 0.89 | 0.81 |
| TRAIN VS. POSEBUSTERS | 0.56 | 0.73 | 0.83 | 0.60 |
| **POCKET SHARED $\geq$ 50** | | | | |
| TRAIN VS. TEST | 0.07 | 0.00 | 0.00 | 0.00 |
| VAL VS. TEST | 0.00 | 0.00 | 0.00 | 0.00 |
| TRAIN VS. VAL | 0.00 | 0.00 | 0.00 | 0.00 |
| TRAIN VS. POSEBUSTERS | 0.57 | 0.42 | 0.81 | 0.51 |
| **PROTEIN LDDT GLOBAL $\geq$ 50** | | | | |
| TRAIN VS. TEST | 0.03 | 0.59 | 0.00 | 0.14 |
| VAL VS. TEST | 0.00 | 0.17 | 0.00 | 0.04 |
| TRAIN VS. VAL | 0.89 | 0.80 | 0.90 | 0.82 |
| TRAIN VS. POSEBUSTERS | 0.56 | 0.75 | 0.81 | 0.60 |
| **PROTEIN SEQSIM $\geq$ 30** | | | | |
| TRAIN VS. TEST | 0.34 | 0.70 | 0.00 | 0.40 |
| VAL VS. TEST | 0.15 | 0.29 | 0.00 | 0.18 |
| TRAIN VS. VAL | 0.93 | 0.84 | 0.96 | 0.85 |
| TRAIN VS. POSEBUSTERS | 0.71 | 0.83 | 0.84 | 0.73 |
| **LIGAND SIMILARITY $\geq$ 30** | | | | |
| TRAIN VS. TEST | 0.66 | 0.70 | 0.68 | 0.69 |
| VAL VS. TEST | 0.62 | 0.65 | 0.64 | 0.63 |
| TRAIN VS. VAL | 0.72 | 0.72 | 0.80 | 0.79 |
| TRAIN VS. POSEBUSTERS | 0.58 | 0.57 | 0.58 | 0.57 |

*Table A11.* Fraction of leaked systems between different dataset-splits for selected similarity metrics and thresholds. For PLINDER-splits we only report on high quality test structures

| METRIC | PDBBIND -ORIGINAL | PDBBIND -LP | PDBBIND -DIFFDOCK | DOCKGEN | PLINDER -ECOD | PLINDER -TIME | PLINDER -PL50 |
|---|---|---|---|---|---|---|---|
| **PLI SHARED ≥ 50** | | | | | | | |
| TRAIN VS. POSEBUSTERS | 0.51 | 0.48 | 0.52 | 0.60 | 0.64 | 0.72 | 0.40 |
| TRAIN VS. TEST | 0.88 | 0.71 | 0.27 | 0.05 | 0.30 | 0.80 | 0.04 |
| TRAIN VS. VAL | 0.73 | 0.64 | 0.78 | 0.13 | 0.05 | 0.78 | 0.10 |
| VAL VS. TEST | 0.89 | 0.69 | 0.08 | 0.01 | 0.05 | 0.59 | 0.00 |
| **POCKET LDDT ≥ 50** | | | | | | | |
| TRAIN VS. POSEBUSTERS | 0.66 | 0.67 | 0.69 | 0.74 | 0.74 | 0.88 | 0.47 |
| TRAIN VS. TEST | 0.97 | 0.84 | 0.53 | 0.09 | 0.49 | 0.96 | 0.00 |
| TRAIN VS. VAL | 0.84 | 0.84 | 0.89 | 0.20 | 0.30 | 0.95 | 0.77 |
| VAL VS. TEST | 0.97 | 0.83 | 0.23 | 0.03 | 0.18 | 0.79 | 0.00 |
| **POCKET SHARED ≥ 50** | | | | | | | |
| TRAIN VS. POSEBUSTERS | 0.63 | 0.64 | 0.65 | 0.70 | 0.70 | 0.83 | 0.47 |
| TRAIN VS. TEST | 0.96 | 0.81 | 0.47 | 0.06 | 0.35 | 0.88 | 0.09 |
| TRAIN VS. VAL | 0.82 | 0.80 | 0.87 | 0.14 | 0.07 | 0.87 | 0.29 |
| VAL VS. TEST | 0.94 | 0.80 | 0.16 | 0.01 | 0.08 | 0.70 | 0.00 |
| **PROTEIN GLOBAL LDDT ≥ 50** | | | | | | | |
| TRAIN VS. POSEBUSTERS | 0.68 | 0.68 | 0.70 | 0.72 | 0.75 | 0.88 | 0.48 |
| TRAIN VS. TEST | 0.97 | 0.85 | 0.53 | 0.10 | 0.49 | 0.95 | 0.01 |
| TRAIN VS. VAL | 0.85 | 0.84 | 0.88 | 0.19 | 0.30 | 0.95 | 0.77 |
| VAL VS. TEST | 0.97 | 0.83 | 0.22 | 0.04 | 0.18 | 0.79 | 0.00 |
| **PROTEIN SEQSIM ≥ 30** | | | | | | | |
| TRAIN VS. POSEBUSTERS | 0.77 | 0.79 | 0.78 | 0.81 | 0.81 | 0.93 | 0.64 |
| TRAIN VS. TEST | 0.97 | 0.90 | 0.58 | 0.19 | 0.60 | 0.98 | 0.37 |
| TRAIN VS. VAL | 0.88 | 0.86 | 0.90 | 0.34 | 0.44 | 0.98 | 0.83 |
| VAL VS. TEST | 0.97 | 0.87 | 0.39 | 0.08 | 0.35 | 0.87 | 0.14 |
| **LIGAND SIMILARITY ≥ 30** | | | | | | | |
| TRAIN VS. POSEBUSTERS | 0.57 | 0.57 | 0.59 | 0.61 | 0.65 | 0.66 | 0.64 |
| TRAIN VS. TEST | 0.55 | 0.40 | 0.37 | 0.53 | 0.52 | 0.54 | 0.58 |
| TRAIN VS. VAL | 0.50 | 0.44 | 0.50 | 0.71 | 0.45 | 0.41 | 0.48 |
| VAL VS. TEST | 0.52 | 0.36 | 0.25 | 0.54 | 0.50 | 0.46 | 0.51 |

| | | TEST | | POSEBUSTERS | |
|---|---|---|---|---|---|
| QUALITY | DATASET-SPLIT | TOP-10 | TOP-1 | TOP-10 | TOP-1 |
| LOW | PLINDER-ECOD | 19.31 ± 0.28 | 13.57 ± 0.09 | - | - |
| | PLINDER-TIME | 37.61 ± 0.10 | 25.35 ± 0.50 | - | - |
| HIGH | PLINDER-ECOD | 26.47 ± 0.39 | 19.02 ± 0.24 | 47.17 ± 0.94 | 38.41 ± 2.11 |
| | PLINDER-PL50 | 25.67 ± 0.61 | 18.19 ± 0.29 | 35.46 ± 2.09 | 29.41 ± 1.46 |
| | PLINDER-TIME | 58.75 ± 0.58 | 45.26 ± 0.38 | 58.16 ± 0.89 | 47.78 ± 0.24 |
| HIGH + LOW | PLINDER-ECOD | 20.85 ± 0.21 | 14.74 ± 0.11 | 47.17 ± 0.94 | 38.41 ± 2.11 |
| | PLINDER-PL50 | 25.67 ± 0.61 | 18.19 ± 0.29 | 35.46 ± 2.09 | 29.41 ± 1.46 |
| | PLINDER-TIME | 41.78 ± 0.16 | 29.27 ± 0.42 | 58.16 ± 0.89 | 47.78 ± 0.24 |
| | PDBBIND-DIFFDOCK (20M) | 47.9 | 35.0 | - | 38.0 |
| | PDBBIND-DIFFDOCK (30M) | 57.0 | 43.0 | - | 50.0 |

*Table A12.* DiffDock performance on varied quality quality test sets reported as percentage success rate of prediction within RMSD <2 Å from reference for Top-1 and Top-10 ranked poses. For each PLINDER split, the trained model is evaluated on its corresponding test set and the PoseBusters set. For on PDBBind-DiffDock split, we show baseline values for the original DiffDock (20M parameters), and DiffDock-L (30M parameters).

*Figure A2.* Characterizing DiffDock PoseBusters benchmark performance relationship with distance cutoffs for different metrics. For this analysis we used an average performance of three models trained on PLINDER splits. The success rate was estimated for any of the top 10 poses being below 2 Å threshold from a reference pose. The effect of "leakage" distance cutoff for various similarity metrics on: (A) the excess success rate observed for "leaked" systems relative to the baseline, (B) Success rate enrichment factor, (C) change in best pose RMSD as a function of "leakage" cutoff (D) cumulative leakage fraction. We note the abrupt jump after 80% distance is due to our choice not to store pairwise similarities below 20% for our PLINDER dataset.

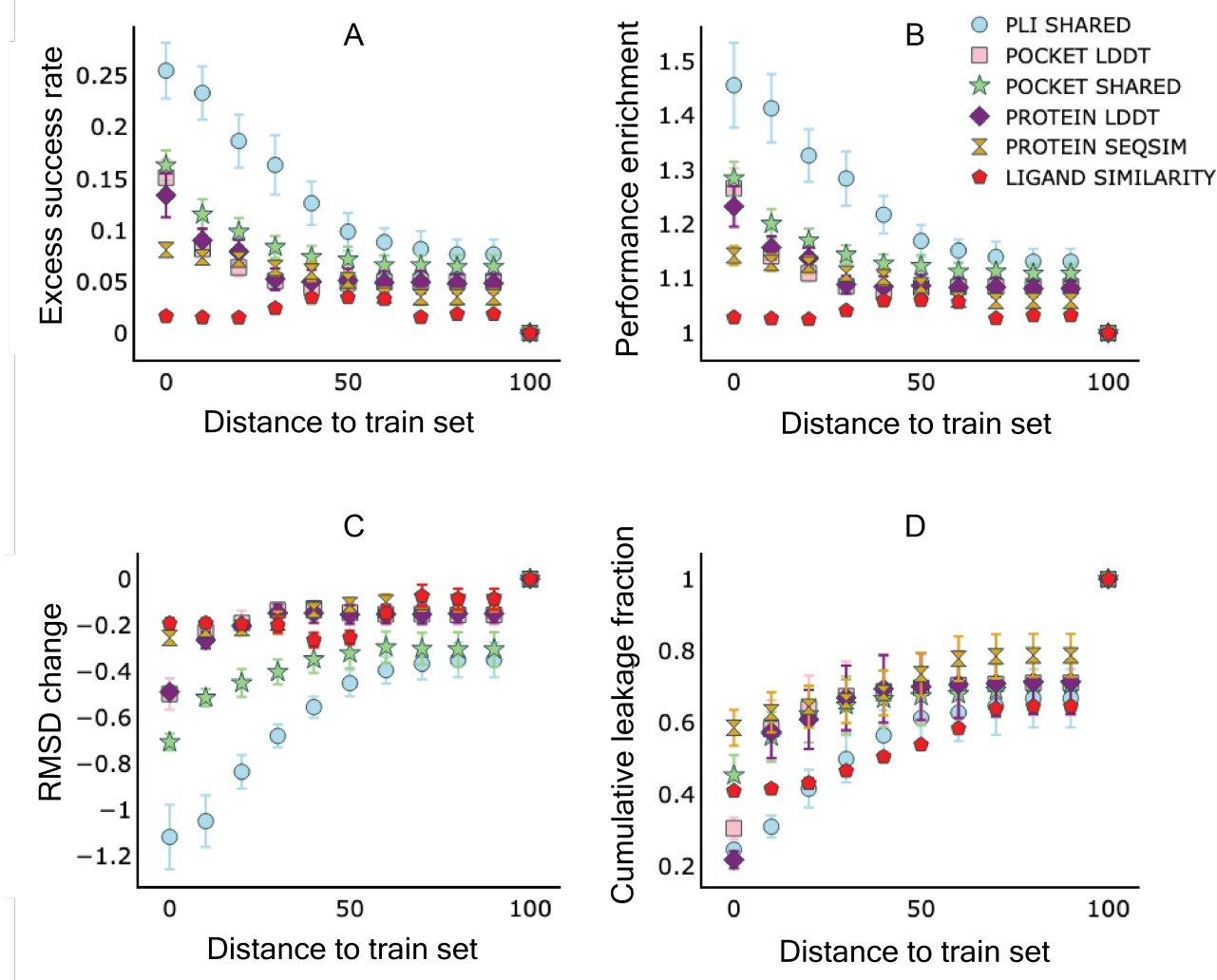

*Table A13.* Hyperparameters used for re-training the DiffDock Score Model. Choices are based on original publication model size and parameters, with the introduction of variable batch sizing.

| Parameter | Option |
|---|---|
| Protein Graph | Coarse-Grain |
| ESM Language Model Embeddings | True |
| Ligand Hydrogens | False |
| Maximum Neighbors in Protein Graph | 24 |
| Receptor Radius | 15 |
| Distance Embedding Method | Sinusoidal |
| Dropout | 0.1 |
| Optimizer | Fused Adam |
| Learning Rate | 0.001 |
| Batch Size | Variable |
| Convolution Layers | 6 |
| Scalar Features | 48 |
| Vector Features | 10 |
| Total Parameters | 20.2M |

*Table A14.* Number of SMILES across the splits that failed to produce ECFP4 fingerprint with RDKit

| CATEGORY | SPLIT | SIZE | NO. FAILED |
|---|---|---|---|
| PLINDER-ECOD | TEST | 11401 | 186 |
| PLINDER-ECOD | TRAIN | 77411 | 757 |
| PLINDER-ECOD | VAL | 10169 | 158 |
| PLINDER-TIME | TEST | 10895 | 141 |
| PLINDER-TIME | TRAIN | 76950 | 804 |
| PLINDER-TIME | VAL | 11392 | 163 |
| PLINDER-PL50 | TEST | 3517 | 12 |
| PLINDER-PL50 | TRAIN | 57734 | 1888 |
| PLINDER-PL50 | VAL | 3459 | 43 |