# OpenReview forum: "PLINDER: The protein-ligand interactions dataset and evaluation resource"
_ICML.cc/2024/Workshop/ML4LMS — ML4LMS Poster_

### Official Review · Reviewer_oA7S · 2024-06-03
**Great Work on Protein-Ligand Datasets with Minor Questions**

**Rating:** 8
**Confidence:** 4

**Review:**

Overall, I really enjoyed this paper and rank is highly. The work presented in this paper is extremely important and impressive. The authors have tackled a significant challenge in the field of protein-ligand interactions (PLI) and have made notable contributions. However, there are several areas where the paper could be strengthened, particularly in terms of visualizations and addressing some key questions regarding the methodology.

Specific Comments and Suggestions:
The paper would greatly benefit from more visualizations. Presenting different datasets visually, as well as providing function-specific examples from the dataset, would help readers better understand the scope and application of the work. These visual aids can also make the paper more accessible and engaging.

Questions and Clarifications:
1. Annotations (Appendix Table A1): Are the annotations listed in Appendix Table A1 present in all the PLI systems? Clarification on this point would help in understanding the consistency and comprehensiveness of the dataset.

2. Similarity Metrics and Dataset Splits: Since similarity metrics between training and test sets can vary widely across different goals or tasks within the protein-ligand field, is there a way for users to tweak these test/train similarity metrics in PLINDER? Or are the different splits pre-determined? Flexibility in this aspect could significantly enhance the utility of the tool.

3. PLI Similarity Metrics:
- Hash Collisions: Is there any possibility that different interactions could result in the same hash? If novel proteins or molecules are added, how does the method ensure that they are accurately distinguished?
- Importance of Interactions: The method assumes that all interactions are equally important, which may not always be the case. Is there a way to incorporate different metrics that might be more important for binding in this method? This could provide a more nuanced and accurate representation of interactions.
- Jaccard Index Assumption: Is there a way to tweak the Jaccard index so that it doesn't make the assumption that the counts of interactions are more important than the presence/absence of an interaction? This modification could offer a more balanced assessment of interaction significance.
- Quality of Pocket Residue Alignment: Can the quality of the alignment of pocket residues be considered in this similarity metric? Incorporating this factor could improve the accuracy of similarity assessments, particularly in cases where alignment quality varies.

In conclusion, this paper presents valuable and impressive work in the field of protein-ligand interactions. Addressing the above points could further enhance its impact and clarity. I recommend accepting the paper with the suggested revisions/considerations for future work.

---

### Official Review · Reviewer_eDoL · 2024-06-12
**PLINDER - Review**

**Rating:** 7
**Confidence:** 4

**Review:**

The paper titled "PLINDER: The Protein-Ligand Interactions Dataset and Resource" presents a comprehensive dataset designed to advance the field of protein-ligand interaction (PLI) prediction, crucial for drug discovery and protein engineering. The authors propose a method to generate training and evaluation splits to minimize task-specific leakage and optimize test set quality.

Strengths: This paper has 4 main strengths, (1) Dataset Size and Diversity, (2) Comprehensive Annotations, (3) Linkage to Apo and Predicted Structures and (4) Evaluation Methodology.

PLINDER is significantly large and diverse, addressing a common limitation in existing datasets which often suffer from size and diversity constraints. Each PLI system in PLINDER is extensively annotated with over 500 features, providing a rich resource for detailed analysis and method development. Linking holo complexes to their corresponding apo and predicted structures facilitates more realistic inference scenarios, which is crucial for evaluating the generalizability of computational methods. The proposed approach for generating training and evaluation splits is well-thought-out, aiming to minimize leakage and maximize the quality of the test sets, which is a critical consideration for fair method assessment.

Weaknesses: This paper has 3 main weaknesses, (1) Implementation Details, (2) Evaluation Metrics and (3) Broader Applicability.

While the paper provides a high-level overview of the dataset and its features, it lacks detailed implementation specifics. For instance, more information on how the annotations were generated and the computational resources required would be beneficial. The paper mentions comparing the performance of DiffDock retrained with different splits but does not delve deeply into the evaluation metrics used for these comparisons. A more detailed analysis of the results, including statistical significance, would strengthen the claims. While the dataset is comprehensive, the paper could further discuss its applicability to different domains beyond drug discovery, such as biotechnology or materials science, to emphasize its broader impact.

Recommendations - 3 main recommendations:

1. Expand Methodology Section:

Include more detailed explanations of the dataset curation process, annotation generation, and the specific algorithms or tools used. Provide additional insights into the computational infrastructure and resources utilized to handle such a large dataset.

2. Detailed Results and Metrics:

Offer a more granular analysis of the evaluation results, including specific metrics and their statistical significance, to provide a clearer picture of the dataset's impact on model performance. Compare the performance of additional models beyond DiffDock to illustrate the dataset's versatility and robustness.

3. Discuss Broader Implications:

Elaborate on potential applications of PLINDER in other fields, highlighting how its diverse and comprehensive nature can benefit various scientific and engineering disciplines. Consider discussing future extensions or updates to the dataset that could address emerging challenges or incorporate new types of data.

Overall Assessment:

This paper presents a significant contribution to the field of protein-ligand interactions by introducing a highly comprehensive and annotated dataset. Despite some areas needing more detail, the dataset's size, diversity, and the thoughtful approach to minimizing information leakage make it a valuable resource for advancing computational methods in drug discovery and beyond.

---

### Official Review · Reviewer_zHh1 · 2024-06-12
**Large protein-ligand dataset with carefully curated splits**

**Rating:** 8
**Confidence:** 3

**Review:**

This paper describes a large dataset of protein-ligand interactions, together with extensive annotations and careful splits to avoid redundancy. This resource should be of significant interest to the community working on protein-ligand interactions (of various kinds).

The dataset splitting strategy is thorough, relying on a variety of similarity metrics between interactions, and reasonably clearly described in the paper. One potential suggestion for improving sensitivity of similarity comparisons in future iterations of the database would be to use  domain annotations from a more comprehensive domain database like The Encylopedia of Domains, allowing for domain-based splits.

Through analysis on retraining of DiffDock, the paper also provides clear evidence that the choice of splitting strategy can have a significant impact on downstream performance metrics.

It would be nice to see more extensive examination of the extent of the increased size of the available data on model performance. Currently it is challenging to disentangle the effects of increasing size and degree of test-set overlap. It might help to focus this analysis on the main proposed splitting strategy, while separately providing results for the time-based and ecod-based splitting strategies to highlight the issues with these alternative strategies.

Similarly, Figure 2C is quite an impactful demonstration of the effect of dataset leakage on performance, but if I understand correctly entangles dataset size and degree of leakage - if there was a way to analyse these factors in a more independent way that could be quite interesting.